# Novel Fabrication Processing of Porous Al_2_O_3_/CaAl_12_O_19_ Membrane by Combining Emulsion, Cement Curing and Tape-Casting Methods

**DOI:** 10.3390/membranes12080747

**Published:** 2022-07-29

**Authors:** Li Wang, Binbin Dong, Zhiyu Min, Li Guan, Xichen Zheng, Qingfeng Wang, Chaofan Yin, Rui Zhang, Feihong Wang, Hamidreza Abadikhah, Xin Xu, Gang Wang, Bo Yuan, Daoyuan Yang

**Affiliations:** 1School of Materials Science and Engineering, Henan Province International Joint Laboratory of Materials for Solar Energy Conversion and Lithium Sodium Based Battery, Luoyang Institute of Science and Technology, Luoyang 471023, China; wangli6433@sina.com (L.W.); dbb023@163.com (X.Z.); 200900100037@lit.edu.cn (Q.W.); ycfwust@163.com (C.Y.); 200900101812@lit.edu.cn (R.Z.); 2School of Material Science and Engineering, Henan Key Laboratory of Aeronautical Materials and Application Technology, Zhengzhou University of Aeronautics, Zhengzhou 450046, China; guan_de@zua.edu.cn; 3Department of Materials Science and Engineering, University of Science and Technology of China, Hefei 230026, China; wangfh@mail.ustc.edu.cn (F.W.); hamid@mail.ustc.edu.cn (H.A.); xuxin@ustc.edu.cn (X.X.); 4State Key Laboratory of Advanced Refractories, Sinosteel Luoyang Institute of Refractories Research Co., Ltd., Luoyang 471039, China; boyuan0401@gmail.com; 5School of Materials Science and Engineering, Zhengzhou University, Zhengzhou 450001, China; brancebob@gmail.com

**Keywords:** porous Al_2_O_3_/CaAl_12_O_19_ membranes, emulsion, cement curing, tape-casting, gas permeability

## Abstract

This paper presented an innovative method for fabrication of a porous Al_2_O_3_/CaAl_12_O_19_ ceramic membrane by combining emulsion method, cement curing and tape-casting technologies. The ceramic membrane featured a smooth surface and a porous internal structure. By adjusting the oil–water volume ratio from 1:1 to 4:1, the porosity of the samples increases from 45.6 to 67.3%, density decreases from 2.07 to 1.32 g/cm^3^ and bending strength decreases from 64.3 ± 1.2 to 31.7 ± 0.6 MPa. More significantly, the membranes showed great gas permeability (1.2 × 10^7^–2.3 × 10^7^ Lm^−2^ h^−1^ bar^−1^), opening up a wide range of applications in the field of gas filtration processes.

## 1. Introduction

Porous ceramic membranes, because of their exceptional mechanical strength and remarkable chemical stability, are excellent candidates for numerous membrane filtration processes, including wastewater treatment, bioreactors, high temperature gas filtration, etc [1,2,3,4,5,6]. However, because of the limitations of conventional pore-forming techniques, most ceramic membranes have relatively low porosity. Therefore, present engineering applications demand membranes with finer pores, higher porosities, and higher bending strength properties [7,8,9].

There are a variety of pore-forming methods to prepare ceramic membranes, including particle sintering [7], sacrificial templates [8], phase inversion [9], foaming [10], and emulsion methods [11]. Due to process limitations, the porosity of membranes prepared by partial sintering method, sacrificial template method, and inverse-phase method is very low, resulting in poor gas permeability. During the foaming process, the wet foam slurry is a thermodynamically unstable system, making it challenging to regulate the membrane pore size [12].

Owing to its unique properties, such as controlled porosity and pore size distribution, water-in-oil emulsion templates are widely used as an improved pore-forming method for the preparation of porous ceramics. The suspended oil droplets that act as pore formers need to be immobilized before evaporating into the pores [13]. In our previous study, a gel casting technique was used to cure the ceramic slurry for fabrication of a porous ceramic membrane. Additionally, the well-known tape-casting method was used to create ceramic membranes of uniform thickness.

The raw materials which are conventionally used in the gel systems, however, are toxic. Herein, it is critical to discover alternative curing techniques in order to approach a non-toxic method for preparation of porous ceramic membranes. Calcium aluminate cement is a material that has already been widely used in industry. The material will undergo a hydration reaction when exposed to water at room temperature to solidify the ceramic slurry. During the high temperature sintering process, the in situ reaction of calcium hexaluminate (CaAl_12_O_19_) with alumina could produce ceramic membrane materials with superior mechanical properties.

Therefore, this effort aims to prepare a porous Al_2_O_3_/CaAl_12_O_19_ ceramic membrane by combining emulsion method, cement curing, and tape-casting technologies. The phase composition, microstructure, bending strength, and gas permeability of the produced membranes were all thoroughly investigated.

## 2. Materials and Experimental

Calcined alumina (d_50_ = 1.1 μm, Almatis, Qingdao, China) and calcium aluminate cement (CAC, Secar71, Kerneos Aluminate Technology Co., Ltd., Tianjin, China) were used as the ceramic raw materials. Ammonium citrate (AR, Shanghai Puzhen Co., Ltd., Shanghai, China) was used as the dispersant. The oil phase of the suspension emulsion is octane, while the emulsifier is Tween 80. Both components were obtained from China National Pharmaceutical Group and are of analytical grade. First, ammonium citrate, water, alumina powder, and Tween 80 were placed in a stirring tank and stirred for 1 h to obtain a suspension. Then, octane with a mass ratio of 25:1 (octane to Tween80) was added to the suspension. After stirring for 1 h, calcium aluminate cement was added to the suspension. The mass ratio of water, alumina powder, and cement were fixed at 3:7:1. After stirring for 20 min, the slurry was poured onto a glass plate with a 1 mm blade gap using the doctor blade method. The green belt was dried at 110 °C for 12 h and heated in air at 1600 °C for 3 h. To adjust the porosity, several water to oil volume ratios of 1:1, 1:2, 1:3, and 1:4 were tested and evaluated.

Phase analysis was performed by X-ray diffraction (XRD, Philips PW 1700, Amsterdam, The Netherlands). The flexural strength was measured by the three-point bending test method. The microstructure was observed by scanning electron microscope (SEM, JEOL JSM-6390LA, Tokyo, Japan). The open porosity of the membranes was determined by the Archimedes method. The pore size distribution of the membranes was determined by the bubble point method [14]. N_2_ permeation measurement through the membrane was conducted using home-made equipment. The membrane was fixed with glue to the base of a male connector, and then installed in a sealed cylinder. N_2_ was injected into the cylinder in order to pass through the membrane by applying various pressures. The N_2_ permeation through the sample was measured by a soap bubble flow meter [15].

## 3. Results and Discussion

Figure 1 presents the X-ray diffraction pattern of the sample sintered at 1600 °C for 3 h. As shown in the figure, the main crystal phase of the sample is α-alumina, while the CaAl_12_O_19_ (CA_6_) phase can be classified as the secondary phase.

The following procedures are carried out [16,17,18] once the calcium aluminate cement is added to the slurry to produce the finalized ceramic membrane:

hydration process:2CA + 11H → C_2_AH_8_ + AH_3_
(1)
dehydration process:3C_2_AH_8_ → 2C_3_AH_6_ + AH_3_ + 9H(2)
4C_3_AH_6_ + 3AH_3_ → C_12_A_7_ + 33H(3)
and the high temperature sintering process:C_12_A_7_ + 5A → 12CA(4)
CA + A → CA_2_(5)
CA_2_ + 4A → CA_6_ (CaAl_12_O_19_)(6)
where C = CaO, A = Al_2_O_3_ and H = H_2_O.

The fracture morphologies of the samples with oil–water volume ratios of 1:1, 2:1, 3:1, and 4:1 are depicted in Figure 2a–d, respectively. Suspensions can be considered as effective pore formers. Under the influence of the emulsifier, the oil droplets become spherical, and after drying and heating, they develop a structure with spherical pores. The pore size is about 5–25 μm, which is attributed to the sufficient emulsification during mechanical stirring. These spherical pores are connected to each other by many fine pores. The discontinuous phases (oil phases) are forced to squeeze closer to one another as the oil concentration increases, which reduces the size of the pores and thins the pore walls. There is adequate space for the oil droplets to disperse in an isolated form when a small amount of oil is added to the suspension. Therefore, as shown in Figure 2a, the inside of the sample contains fewer interconnected pores. As the oil content increases, the suspension contains more spherical oil droplets leading to the generation of higher numbers of interconnected pores and thinner pore walls formation (Figure 2b–d). The network of interconnected pores acts as the main channel to improve the air and water permeability of the membrane. Because the surface of the sample is easier to rearrange and shrinks more during drying and sintering processes, the size of the pores on the surface of the sample is slightly smaller than those inside the membrane. Figure 2e shows the surface morphology of the sample with an oil–water volume ratio of 2:1. The surface of the sample is smooth, which is characteristic of the casting process. A small amount of the emulsion collapsed during the drying process, resulting in a small number of indented pores on the surface of the sample. Consistent with the aforementioned fracture morphology, the surface of the sample has a significant number of connected pores due to the connection between the pore walls, which also serves to improve air and water permeability. Figure 2f shows the cross-sectional morphology of the prepared membrane and confirms the successful formation of a porous Al_2_O_3_/CaAl_12_O_19_ membrane.

In addition, the increase in oil content increased porosity from 45.6 to 67.3%, decreased density from 2.07 to 1.32 g/cm^3^, and decreased bending strength from 64.3 ± 1.2 to 31.7 ± 0.6 MPa (Figure 3). The oil phase served as the pore-forming agent in this investigation, and since the proportions of alumina powder and water were kept constant, higher numbers of interconnected pores were generated by the increase of oil content, leading to higher porosity and thinner pore walls formation. The thinner pore walls facilitate the pores’ connectivity, increase the pore size distribution, and reduce the mechanical strength of the ceramic skeleton. The formation of CaAl_12_O_19_ through high temperature sintering and in situ bridging is beneficial to improve the mechanical strength of the material. The flexural strength of the samples is high and can withstand the high shear force during the filtration process.

Figure 4a shows the pore size distribution of the sintered samples. With the increase of oil content, the average pore size increases from 1.7 to 2.4 μm. In comparison with the large interior holes with sizes ranging from 5 to 25 µm, the sizes of the interconnected holes that play a critical function in separation efficiency of the membrane are smaller, which benefits its application in water treatment. The results of N_2_ permeability of the sample are shown in Figure 4b. The gas permeability of the samples increased significantly (from 1.2 × 10^6^ to 2.3 × 10^6^ Lm^−2^ h^−1^ at the transmembrane pressure of 1 bar) as the oil content increased, which is compatible with the related structures in the literature.

## 4. Conclusions

The present paper reported a novel method for fabrication of a porous Al_2_O_3_/CaAl_12_O_19_ membrane by emulsion template preparation, cement curing and tape-casting process. The ceramic membrane revealed a smooth surface and a porous internal structure. The increase of oil–water ratio from 1:1 to 4:1 could increase the formation of the interconnected pores, improve the porosity of the membrane, and enhance the permeability properties of the membrane. Higher oil phase content also increased the pore size distribution of the membrane from 1.7 to 2.4 μm. Together with the possibility of large-scale production, the aforementioned outcomes illustrate the commercialization prospects of the fabricated structures in the field of membrane separation technology.

## Figures and Tables

**Figure 1 membranes-12-00747-f001:**
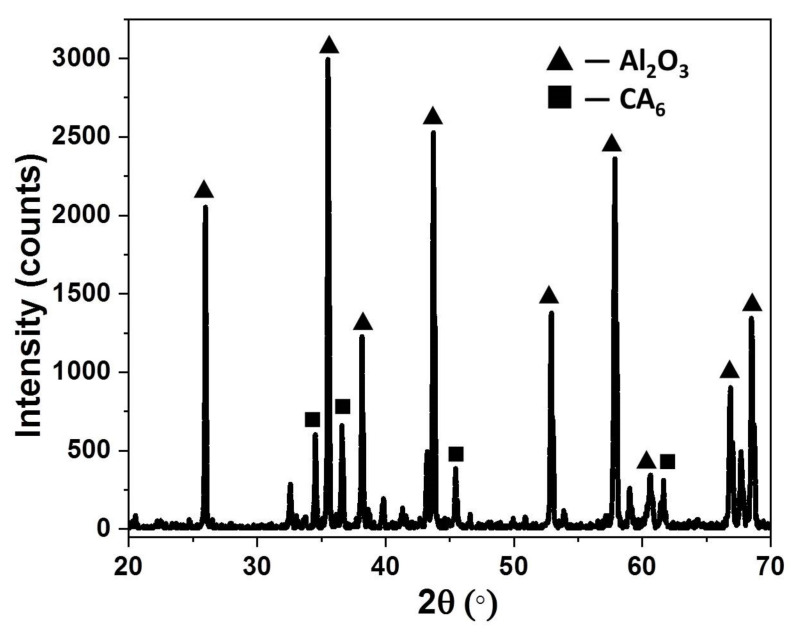
Phase composition of the sintering sample.

**Figure 2 membranes-12-00747-f002:**
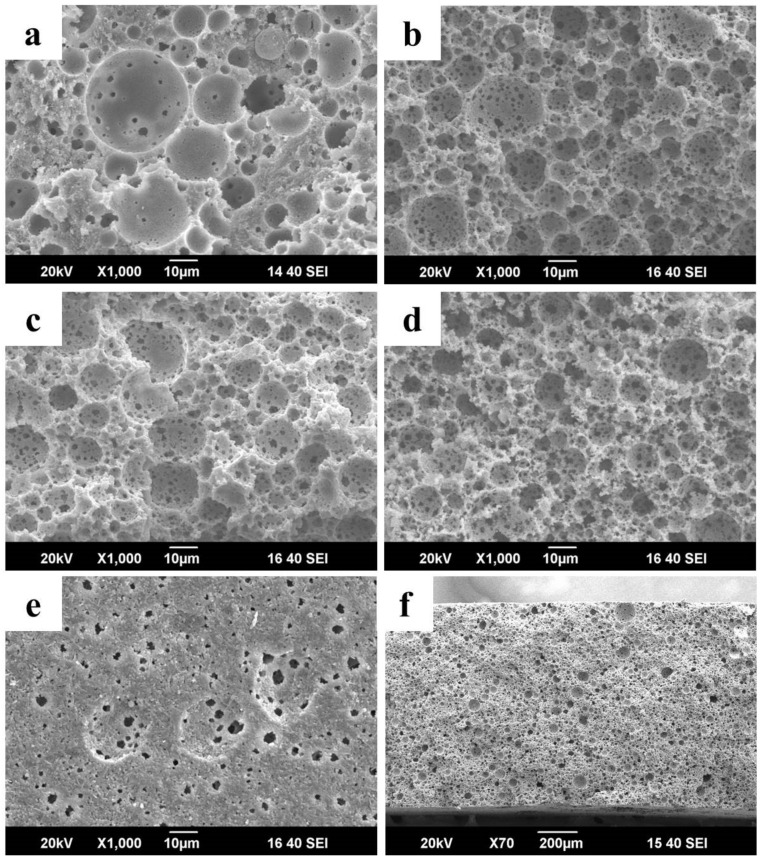
Fracture morphology of the samples with oil–water volume ratio of (**a**) 1:1, (**b**) 2:1, (**c**) 3:1 and (**d**) 4:1, the surface (**e**) and fracture morphology (**f**) of the sample with oil–water volume ratio of 2:1.

**Figure 3 membranes-12-00747-f003:**
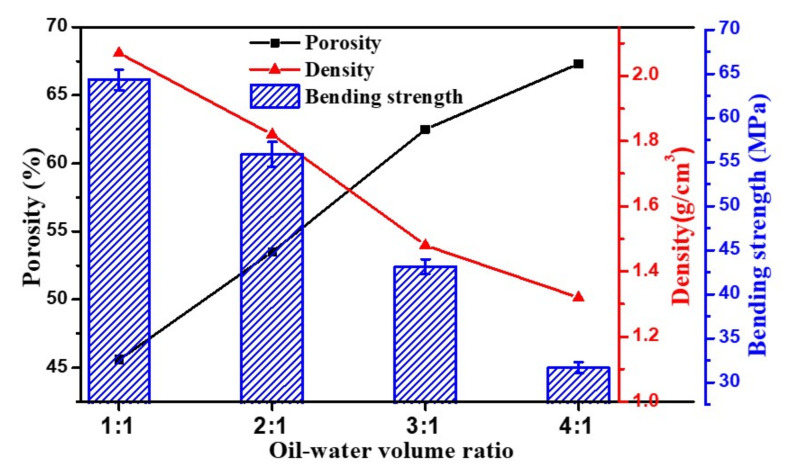
Porosity, density, and bending strength of sintered samples with oil–water volume ratios of 1:1, 2:1, 3:1, and 4:1.

**Figure 4 membranes-12-00747-f004:**
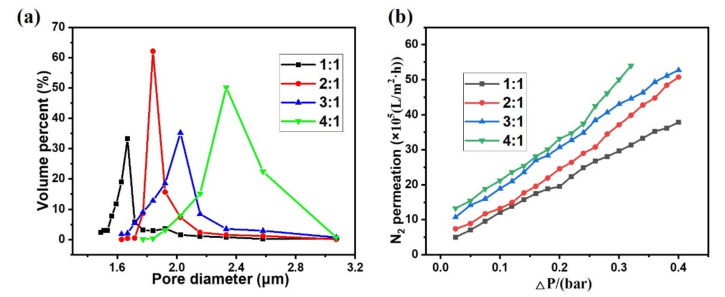
Pore size distribution (**a**) and N_2_ permeability (**b**) of sintered samples with oil–water volume ratios of 1:1, 2:1, 3:1, and 4:1.

## Data Availability

Not applicable.

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
