# Peer review of "Novel Fabrication Processing of Porous Al2O3/CaAl12O19 Membrane by Combining Emulsion, Cement Curing and Tape-Casting Methods"

_membranes, 2022, doi:10.3390/membranes12080747_

Round 1

Reviewer 1 Report

1. The purposes and objectives of this study should be clearly addressed with an appropriate new approaches. This should be provide for better understanding why this research is significant to the readers.

2. The development progress of ceramic membranes with relevant researches should be explained with an appropriate additional references. 

(e.g.,)

Membrane and Water Treatment, 2020, 11(1): 31-39

3. The detailed methods should be inserted, for example, pore diameter measurements and the analytical analysis (N2 flux)

4. How did you measure the membrane surface? The authors mentioned that the ceramic membrane revealed a smooth surface and a porous internal structure

Author Response

Thank you very much for your valuable comments on our present research work. As a result of your recommendation, new information was included in the revised manuscript, as explained in the following response to your comments.

  1. The purposes and objectives of this study should be clearly addressed with an appropriate new approach. This should be provided for better understanding why this research is significant to the readers.

The innovation of the present paper can be highlighted by the presentation of a novel method for fabrication of a porous Al2O3/CaAl12O19 ceramic membrane by combining emulsion method, cement curing and tape casting technologies. Water-in-oil emulsion templates have been widely used as an improved pore-forming method for the preparation of porous ceramics. Calcium aluminate cement is a material that has already been widely utilized in industry. Moreover, tape-casting is a well-established technique to produce flat ceramic structures with tailored thickness. Therefore, in this paper, through the combination of  emulsion method, cement curing and tape casting technologies, a porous Al2O3/CaAl12O19 ceramic membrane could be successfully prepared. (Page 3, line 11-12)

  1. The development progress of ceramic membranes with relevant researches should be explained with an appropriate additional reference. (e.g.,) Membrane and Water Treatment, 2020, 11(1): 31-39

       The development progress of ceramic membranes with relevant researches has been explained with appropriate additional references in the revised manuscript. (References [1-2])

  1. The detailed methods should be inserted, for example, pore diameter measurements and the analytical analysis (N2 flux)

The pore size distribution of the membranes was determined by the bubble point method [14]. N2 permeation measurement through the membrane was conducted using a home-made equipment. The membrane was fixed by glue on the base of a male connector, and then covered by a cylinder. N2 was injected into the cylinder in order to pass through the membrane at different applied pressures. The N2 permeation through the sample was measured by a soap bubble flow meter.

  1. How did you measure the membrane surface? The authors mentioned that the ceramic membrane revealed a smooth surface and a porous internal structure.

The smooth surface of the membranes was observed and analyzed using scanning microscopy analysis, as shown in Figure 2e and 2f.  In fact, one of the advantages of tape casting method is to produce the membranes with smooth surfaces. It is worthy to note that, after combination of tape casting method with the cement curing and emulsion pore-forming techniques, the membranes still revealed a smooth surface structure. It is beneficial for assembling membrane components for different applications.

Reviewer 2 Report

The authors fabricated  a porous Al2O3/CaAl12O19 membrane by emulsion template preparation, cement curing and tape casting process. The ceramic membrane showed a smooth surface and a porous internal structure (1.7μm~2.4μm)  membrane.  However, they have not tested any application of this pore size. The reviewer  is of opinion that the authors can test the similar pore size solvent or any material separation in the field of separation technology. 
